# Implicit Semi-auto-regressive Image-to-Video Diffusion

## Abstract

Diffusion models have demonstrated exceptional performance in various generative domains, particularly in the context of image and video generation. Despite their remarkable success, image-to-video (I2V) generation still remains a formidable challenge for most existing methods. Prior research has primarily concentrated on temporally modeling the entire video sequence, resulting in semantic correspondence but often lacking consistency with the initial image input in detail. In this paper, we present a novel temporal recurrent look-back approach for modeling video dynamics, leveraging prior information from the first frame (provided as a given image) as an implicit semi-auto-regressive process. Conditioned solely on preceding frames, our approach achieves enhanced consistency with the initial frame, thus avoiding unexpected generation results. Furthermore, we introduce a hybrid input initialization strategy to enhance the propagation of information within the look-back module. Our extensive experiments demonstrate that our approach is able to generate video clips with greater detail consistency relative to the provided image.

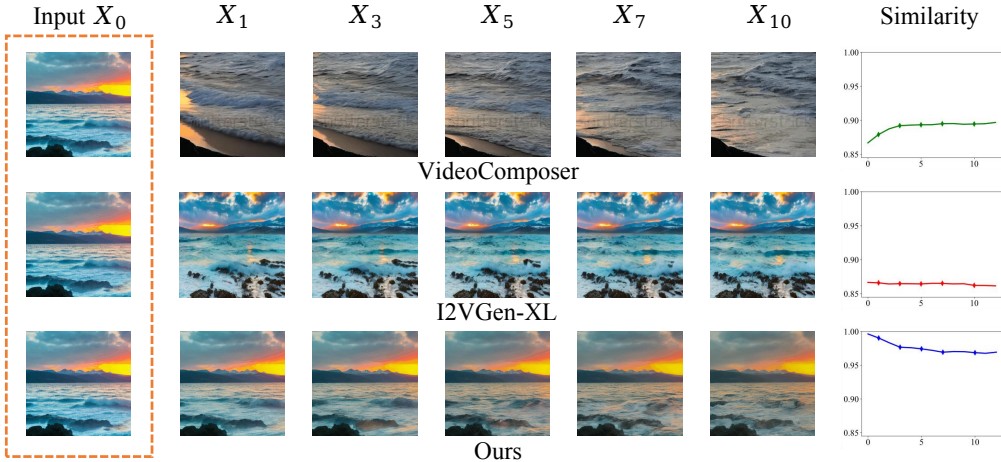

Figure 1: Visualization of similarity between video frames and first frame condition. Our method keeps much higher consistency with given image than other SOTA methods.

## 1 Introduction

Diffusion-based video generation has delivered promising results across diverse applications over the past year. This includes text-to-video (T2V) synthesis (Ho et al., 2022b; Blattmann et al., 2023; Singer et al., 2022; Zhou et al., 2022), image-to-video (I2V) generation (Ni et al., 2023; Wang et al., 2023), and video editing/translation (Ouyang et al., 2023; Chai et al., 2023), among other domains. Make-A-Video (Singer et al., 2022) tackled diffusion-based T2V tasks by incorporating a temporal attention module into the diffusion U-Net, originally from the transformer-based CogVideo (Hong et al., 2022). MagicVideo (Zhou et al., 2022) employs temporal attention in latent space, offering computational

efficiency benefits. Noteworthy alternatives like Latent-Shift (An et al., 2023) and SimDA (Xing et al., 2023) utilized temporal shift modules and adapters to enhance efficiency. The outstanding performance of several noteworthy works further underscores that global temporal attention serves as a promising paradigm in the T2V domain (Guo et al., 2023; Tang et al., 2023b; Wang et al., 2023).

Image-to-video generation, a less-explored domain, presents a more realistic yet challenging task for diffusion models. In essence, I2V aims to generate a video sequence that maintains somehow relevance to the provided image. MCVD (Voleti et al., 2022) and RaMViD (Höppe et al., 2022) represent pioneering efforts in diffusion-based I2V. They utilize temporal masks to facilitate video prediction and infilling, thus also have the ability of I2V generation. Based on the success of the T2V, there are also some multi-modality methods (Tang et al., 2023b; Wang et al., 2023) which claims to handle I2V task. While these methods showcase remarkable results across various tasks, they often treat images similarly to text prompts, globally injecting them into the model and relying on temporal attention for consistency. This approach, however, can result in the loss of fine-grained visual details, achieving only semantic correspondence with the given image.

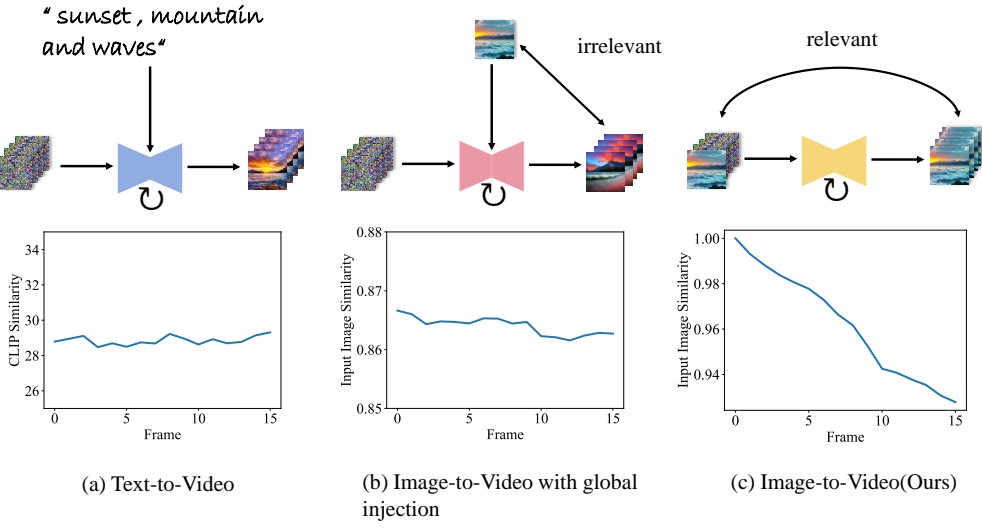

Figure 2: Similarity with input condition

Considering the significant differences between T2V and I2V, it is necessary to reconsider the reasonability of directly leveraging the global condition injection and global temporal attention to I2V task. Typically, there are two major settings for I2V generation. In the first setting, commonly employed by most multi-modality methods (Wang et al., 2023; Tang et al., 2023b), there is no explicit or implicit constraints on fine-grained visual similarity between the generated video and the given image. The image condition is frame-wise injected into the model. Combined with temporal attention, their results are often locally well-correlated, but they may fail to closely align with the given image. Another setting places a strong emphasis on ensuring strict consistency with the provided image (Voleti et al., 2022; Höppe et al., 2022; Nikankin et al., 2022). In this setting, it is a strong assumption that the first frame should be the same as the given image. However, when temporal attention is employed, it often leads to noticeable jumps and flickers between the first and later frames because the information from each frame gets partially blended with subsequent frames. In this paper, we opt for the latter setting, as it is more practical and valuable in real-world applications.

In this work, we propose a semi-auto-regressive approach for I2V generation. In our method, each frame in the video sequence only condition on its previous frame. We design a look-back convolution layer to propagate information from the previous frame and update the U-Net hidden representation frame-by-frame. To reduce inconsistency with the given image when generating iteratively, we initialize the first frame of the generated video using the given image. Other frames are initialized with Gaussian noise. Combining this hybrid initialization and look-back modules, the visual consistency of the generated videos and given images are largely improved compared to

previous works (Tang et al., 2023b; Luo et al., 2023; Wang et al., 2023; Ni et al., 2023). We also design a simple metric to visualize the consistency and relevance with the given image, i.e. Input Similarity(ISIM). We draw the curve of this metric to evaluate different I2V methods.

Our contributions can be summarized as follows:

- We propose a novel implicit semi-auto-regressive approach for diffusion-based image-to-video synthesis. It improves both temporal consistency and relevance to the given image.

- Specifically, we design a look-back recurrent convolutional module inserted to diffusion U-Net to implement this framework. It provides coherent generation results with high computation efficiency.

- We introduce a hybrid input initialization strategy for training and inference to enhance information propagation of the look-back module.

## 2 METHODOLOGY

In this section, we begin by providing a formal definition of the problem. Subsequently, we offer an overview of our method, as illustrated in Fig. 3. Next, we delve into the intricacies of the carefully designed look-back module and semi-auto-regressive generation method. Finally, we introduce our hybrid image-noise initialization strategy, together with training and sampling methods.

### 2.1 PROBLEM DEFINITION

As mentioned above, we aim to generate a video that is consistent with the given image and resembles real videos. To simplify this problem, we assume the first frame is the given image. Therefore, our objective can be formulated as follows,

$$L_\theta = D_{\text{KL}} \left( p_{\text{data}}(x^{1:T}|\hat{x}^0) \parallel p_\theta(x_{1:T}|\hat{x}^0) \right), \tag{1}$$

where $\hat{x}^0$ represents the given image as a condition, $x^{1:T}$ denotes generated frame sequences. $p_{\text{data}}(\cdot)$ represents the realistic distribution of video data sequence, where $p_\theta(\cdot)$ serves as frames distribution of generated videos. We minimize the Kullback-Leibler (KL) divergence between these two distributions using diffusion models to generate realistic videos.

### 2.2 SEMI-AUTO-REGRESSIVE VIDEO GENERATION

We propose implicit semi-auto-regressive image-to-video diffusion, which aims to generate relevant video sequence given an initial frame. As shown in Fig. 3, our method mainly consists of a temporary recurrent look-back module. To enhance the temporal consistency from coarse to fine level, we append a look-back module to capture temporal motion after each layer of diffusion U-Net. At the inference stage, we concatenate the given image $\hat{x}_0$ with $L-1$ Gaussian noise as the input of diffusion U-Net. In each look-back module, we slide the look-back kernel along the time axis from the first to the last frame, updating each frame's hidden states in the U-Net iteratively. To enhance global smoothness, a lightweight 3D normal convolution layer is added after each look-back module as an adapter.

**Look-back sliding window** Inspired by the great success of the next token prediction task in large language models, we find the I2V task can also be modeled as a similar process. Specifically, we generate the next frame by jointly considering information from its previous frames and its own representations.

Since video generation is more computation-intensive than token generation, it is more desirable to only condition each frame on several preceding frames and its own hidden features instead of all the preceding frames. This would avoid the quadratic time complexity of generating with global attention and the linear space complexity which could limit the length of video generation.

Hence, we devise a local convolutional layer that functions as a sliding window, called look-back convolution. In each iteration, it recurrently updates the U-Net hidden states frame by frame through

convolutional operations involving both the preceding frames and its own hidden state, as depicted in Fig. 3

When the look-back window slides, its input is composed of frames that were just updated. Therefore, within a single look-back layer, information can be propagated from the first frame to the last frame. This marks a significant difference between traditional convolutions and causal convolutions. The look-back module forms a receptive field of all preceding frames, enabling information to flow unidirectionally from beginning to end.

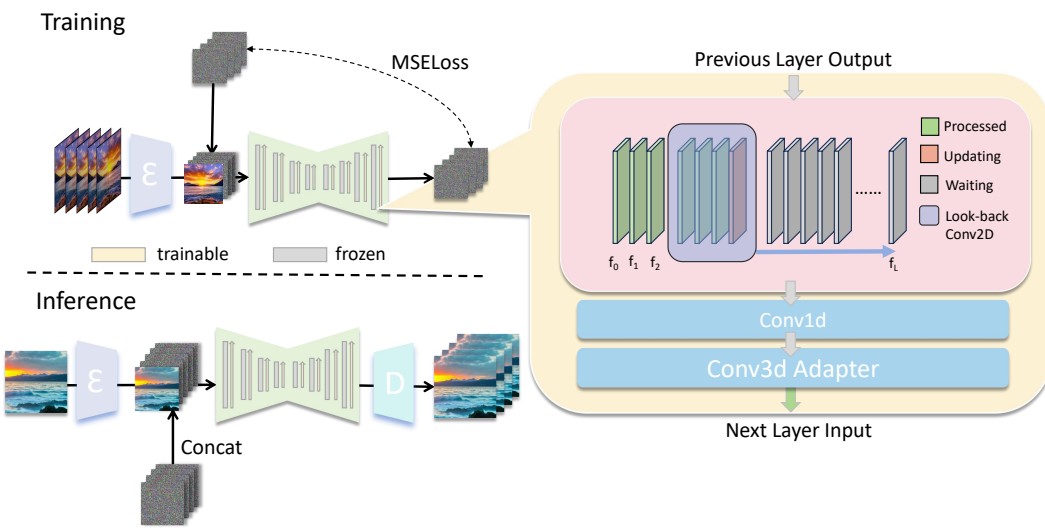

Figure 3: (a) Pipeline of our framework. (b) The temporal look-back module

**Implementation Details of look-back module** In our implementation, each look-back convolution layer is configured as a 2D convolutional layer. It accepts input from a number of channels equal to the product of the look-back depth and the number of channels in the hidden states. The output from this layer consists of channels equal to the number of hidden states. We further integrate a subsequent 1D convolution layer after the look-back layer to refine hidden representation in channel level.

Moreover, similar to SimDA (Xing et al., 2023), we also introduce a lightweight depth-wise normal 3D convolution layer in the entire hidden states of inflated 3D U-Net after each look-back convolution layer. Its purpose is to enhance temporal consistency. The look-back module can be summarized as Algorithm 1.

---

**Algorithm 1:** Look-back Module

**Input:** Previous layer hidden states $h_{\text{in}}^{0:L}$, channel number $n$, look-back depth $d$
**Output:** Updated hidden states $h_{\text{out}}^{0:L}$
Initialize Conv2d input channels $d \cdot n$, output channels $n$
$h_{\text{out}}^{0:L} = h_{\text{in}}^{0:L}$
**for** $i = 0, 1, \cdots, L$ **do**
$\quad | \quad h_{\text{out}}^i = \text{Conv2d}(h_{\text{out}}^{i-d+1:i+1})$
**end**
$h_{\text{out}}^{0:L} = \text{Adapter}(\text{Conv1d}(h_{\text{out}}^{0:L}))$
**return** $h_{out}^{0:L}$

---

## 2.3 Hybird Image-noise Initialization

Under semi-auto-regressive scenario, we use a hybrid image-noise input inspired by RaMViD (Höppe et al., 2022). In our ablation study 7, other initialization options are explored. We set the original given image as the first frame of the input tensor in both the training stage and inference stage. The subsequent frames undergo a process of noise addition during training, followed by prediction, while during sampling, they are denoised from random noise. Refer to Algorithms 2 and 3 for detailed insights into our Training and Sampling procedures

**Training** Similar to original DDPM training objectives, we also employ DDPM forward and predict noise, calculating MSE loss. Note that when training, only frames $x^{1:L}$ are used to calculate loss. The objective of training can be formulated as

$$\mathcal{L} = \mathbb{E}_{x_0^{0:L} \sim p_{\text{data}}(x^{0:L}), \epsilon^{1:L} \sim N(0,I)} \left[ \left\| \epsilon^{1:L} - \epsilon_\theta(\{x_0^0, \sqrt{\bar{\alpha}_t} x^{1:L} + \sqrt{1 - \bar{\alpha}_t} \epsilon^{1:L}\}, t)^{1:L} \right\|_2 \right] \quad (2)$$

**Inference** At the inference stage, the model just serves as a video denoiser with DDIM (Song et al., 2020) sampling strategy. All frames except the first frame are sampled from noise, while we preserve the first frame stable at each timestep same as the input.

| **Algorithm 2:** I2V Training | **Algorithm 3:** I2V Inference |
|---|---|
| **while** *not converged* **do** | Get first frame input $\hat{x}^0$ |
| $\quad$ Sample $x_0^{0:L} \sim p_{\text{data}}(x^{0:L})$ | $x_T^{1:L} \sim N(0, I)$ |
| $\quad \epsilon^{1:L} \sim N(0, I)$ | $x_T^{0:L} = \text{Concat}(\hat{x}^0, x_T^{1:L})$ |
| $\quad t \sim \mathcal{U}(0, 1, 2, .., T)$ | **for** $t = T, \cdots, 1$ **do** |
| $\quad x_t^{1:L} = \sqrt{\bar{\alpha}_t} x^{1:L} + \sqrt{1 - \bar{\alpha}_t} \epsilon^{1:L}$ | $\quad x_{t-1}^{1:L} = x_t^{1:L} - \frac{\beta_t}{\sqrt{1-\bar{\alpha}_t}} \epsilon_\theta(x_t^{0:L}, t)^{1:L}$ |
| $\quad x_{\text{in}} = \text{Concat}(x_0^0, x_t^{1:T})$ | $\quad x_{t-1}^0 = \hat{x}^0$ |
| $\quad \theta = \theta - \nabla_\theta \|\epsilon^{1:L} - \epsilon_\theta(x_{\text{input}}, t)^{1:L}\|$ | **end** |
| **end** | **return** $x_0^{0:L}$ |
| **return** $\theta$ | |

**The explanation of hybrid input** We further argue that such a hybrid input force model to learn the special representations of the initial frame. Compared to methods utilizing DDIM Inversion or simply adding noise for initialization during inference, starting with a noise-free initial frame could extract higher-quality representation (Tang et al., 2023a). This work also posits that well-trained T2I models excel as image feature extractors. Thus with the help of the look-back module, information can propagate along the time axis more efficiently, with less interference from random noise.

## 3 Experiments

**Experiments Setup** We utilized Stable Diffusion v1.5 (Rombach et al., 2021) to initialize the 2D U-Net backbone for our experiments. The model was trained on a 5000 clips subset of WebVid-2M (Bain et al., 2021). To optimize training efficiency, we center-cropped and resized the video clips to dimensions of $256 \times 256$. We employed a frame sampling strategy with a stride of 4, and the total sequence length was set to 32 frames. In our implementation, the look-back module features a kernel size of $5 \times 5 \times 5$, i.e. the look-back depth is set at 5, and the Conv2d kernel size is $5 \times 5$. Simultaneously, the depth-wise 3D Adapter employs a kernel size of $3 \times 3 \times 3$. Our selected learning rate for the training process is 1e-4. The entire training process encompassed 150k steps, with a batch size of 4 and a gradient accumulation step of 4. The training took roughly 72 hours on a single Nvidia A100 80G GPU. It is worth noting that further training may lead to improved generation results.

### 3.1 Qualitative Results

In this section, we present examples of the results obtained through our generation process, as illustrated in Figure 4. We generated 16-frame videos using a specified image at a resolution of $256 \times 256$. It's important to note that all initial images were generated by Stable Diffusion v1.5 using a given prompt. This same prompt was also input into our model to enhance the quality of the generated content.

To assess the performance of our model, we conducted a comparative analysis with other generic I2V diffusion models. The results of this comparison are depicted in Figure 5. Our experiments clearly demonstrate that existing methods struggle to produce videos that are relevant to the input image, as discussed in our introduction.

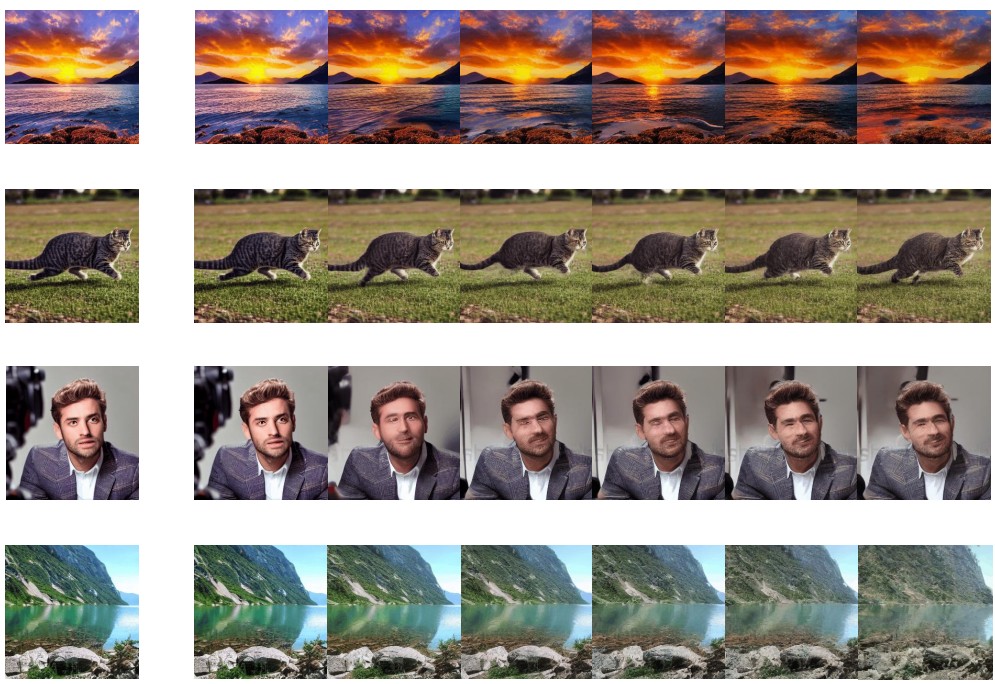

Figure 4: I2V Generation Results. The leftmost one is the given image.

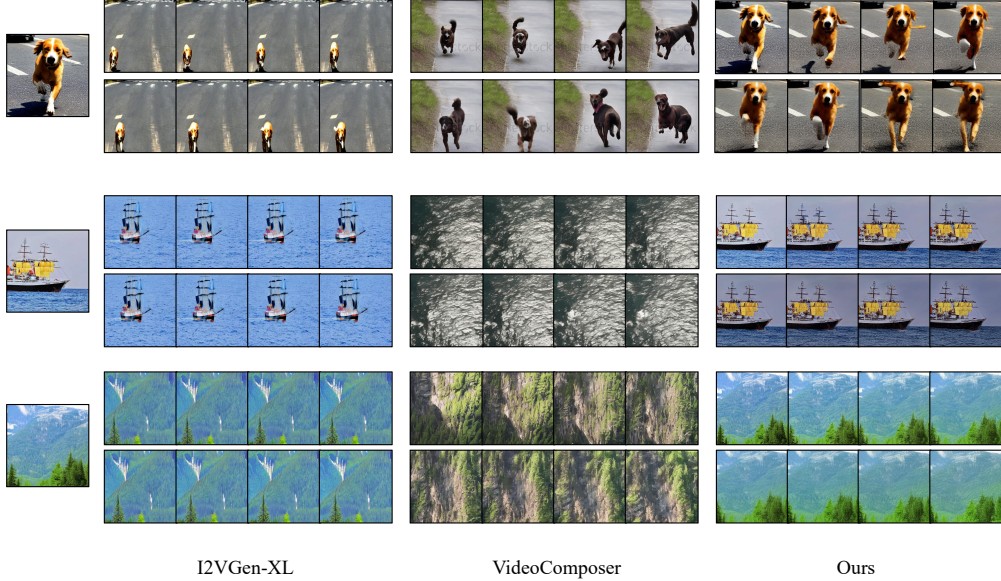

I2VGen-XL                    VideoComposer                    Ours

Figure 5: We present three representative examples of three models. VideoComposer (Wang et al., 2023) and I2VGen-XL (Luo et al., 2023; Wang et al., 2023) generates high realistic videos but loss most of the input details.

## 3.2 QUANTITIVE RESULTS

For I2V task, we assess several metrics, including the cosine similarity between the first frame and the input image (ISIM(1st)), the average cosine similarity with the given image (ISIM(Avg)), and the ISIM Decay Rate. Additionally, we calculate the average CLIP Similarity (CLIPSIM) when a text prompt is provided. We report these metrics for generic I2V diffusion approaches, specifically VideoComposer (Wang et al., 2023) and I2VGen-XL (Luo et al., 2023; Wang et al., 2023). Furthermore, we provide these metrics based on ground-truth data from WebVid (Bain et al., 2021). The results are presented in Table 1.

Table 1: Quantitative Evaluation Results. We assess the performance of our method under two settings: with and without a text prompt. The presence of a text prompt boosts generation quality.

| Model | ISIM(1st)↑ | ISIM(Avg) ↑ | CLIPSIM↑ | ISIM Decay |
|---|---|---|---|---|
| Ground-truth Data | 1.00 | 0.87 | 32.0 | 0.012 |
| VideoComposer(I2V) | 0.85 | 0.82 | 28.3 | 0.0039 |
| I2VGen-XL(I2V) | 0.88 | 0.85 | - | 0.0033 |
| **Ours(I2V)** w/o prompt | **1.00** | **0.94** | - | **0.0062** |
| **Ours(I2V)** w/ prompt | **1.00** | **0.96** | **30.2** | **0.0041** |

## 3.3 INFERENCE EFFICIENCY

The recurrent nature of our approach presents a challenge for hardware when performing parallel inference in a video sequence. However, our methods overcome this limitation by achieving linear time complexity, thanks to the utilization of a sliding window. Experimental results demonstrate the effectiveness of our methods in generating long videos within a reasonable time and memory usage, as illustrated in Figure 6.

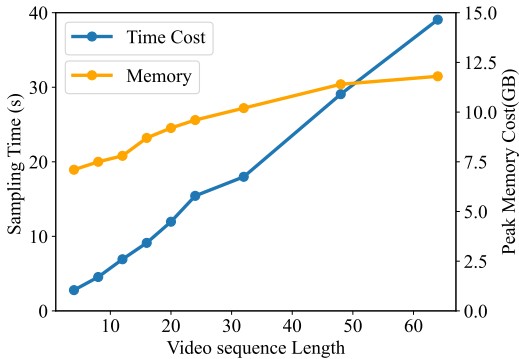

Figure 6: Inference Time and Memory Cost. Our method is able to generate 48 frames video on a personal GPU such as NVIDIA RTX3060 with extremely limited memory source. The depicted time cost in the image corresponds to a single A100 40G GPU.

## 3.4 ABLATION STUDY

**Effect of Look-back Module** We highlight the distinction between conventional 3D convolution and our innovative look-back convolution. When used in isolation, normal 3D convolution tends to produce smooth yet unsatisfactory results, while look-back convolutions achieve greater consistency with the first frame, albeit at the cost of some smoothness. However, when these two techniques are combined together, they are able to generate exceptional videos that closely align with the provided image.

We also provide generation results in situations where look-back modules are replaced with global attention/masked attention in Appendix A.2.

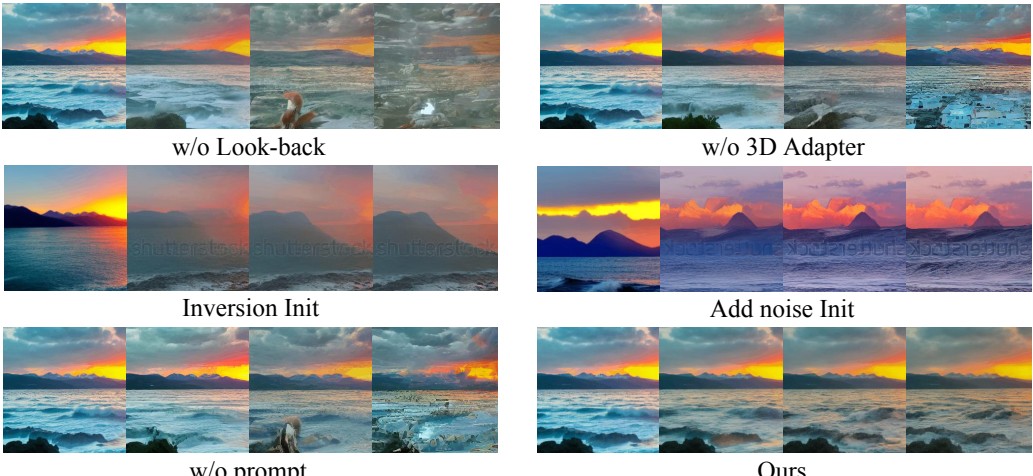

Figure 7: Ablation Study

**Effect of different hybrid strategy** In our evaluation, we also assess the effectiveness of the hybrid strategy incorporated into our model. In addition to our proposed strategy, we compare it with two alternative strategies:

  a. **Add noise** In this approach, we employ the original DDPM training. During inference, we initialize $\sqrt{\bar{\alpha}_T}\hat{x}^0 + \sqrt{1-\bar{\alpha}_T}\epsilon$ as the first frame $x^0$. At each inference step, we forcefully update $x_t^0 = \sqrt{\bar{\alpha}_t}\hat{x}^0 + \sqrt{1-\bar{\alpha}_t}\epsilon$

  b. **Inversion** Similarly, we use the original DDPM training. During inference, we employ DDIM inversion to obtain the noise representation of the first frame. At each look-back module, we forcefully keep the output the same as the input at the position of the first frame, i.e. $h_{out}^0 = h_{in}^0$.

Experimental results in Figure. 7 indicate that Strategy a. fails to generate relevant frames. The generated frames tend to be similar to each other across $x^{1:L}$, but do not exhibit a clear correspondence with $\hat{x}^0$. This result may be attributed to the irrelevant random noise added, which makes the model fail to propagate valuable information.

On the other hand, Strategy b. demonstrates the capacity to generate relevant frames, but the quality is not as good as our proposed strategy. Since diffusion models suffer from extracting fine-grained geometric and semantic features at a high noise level (Tang et al., 2023a), the information propagation within early timesteps during inference is adversely affected, leading to a missed opportunity to propagate valuable information early on. Furthermore, it's worth noting that neither of these two methods achieves a high correspondence with the given image, even for the first frame, primarily due to the non-lossless nature of inversion and noise addition processes.

**Text involvement when inference** We have observed that incorporating text as an additional input for I2V will boost generation quality, particularly far from the video beginning. This phenomenon is illustrated in Figure. 7 Given that our look-back module operates independently of text, we hypothesize that the improved quality is due to the text's role in the basic T2I Model. The integration of cross-attention and classifier-free guidance appears to facilitate the model in generating higher-quality images, thus also producing better video frames. Cross-attention also helps to preserve important information when propagating through the look-back module.

## 4 RELATED WORKS

**Video Generation with Diffusion Models** The pioneering work on video generation using diffusion models can be traced back to RVD (Yang et al., 2022) and VDM (Ho et al., 2022b). Imagen (Ho

et al., 2022a) has adopted a cascaded sampling pipeline and spatial-temporal super-resolution techniques. MCVD (Voleti et al., 2022) utilizes a mask prediction method for video prediction and interpolation. Make-A-Video (Singer et al., 2022) introduces a temporal attention mechanism, while MagicVideo (Zhou et al., 2022) and LVDM (He et al., 2022) incorporate it into the latent space. Tune-A-Video (Wu et al., 2022) and SinFusion (Nikankin et al., 2022) train video diffusion on a single video. Text2Video-Zero (Khachatryan et al., 2023) proposes a zero-shot approach. For enhanced controllability, ControlVideo (Zhang et al., 2023b) and Control-A-Video (Chen et al., 2023) leverage ControlNet (Zhang & Agrawala, 2023). DreamPose (Karras et al., 2023), Follow Your Pose (Ma et al., 2023), and MagicAvatar (Zhang et al., 2023a) are specially designed for human pose-guided generation. More recently, AnimateDiff (Guo et al., 2023) enhances video diffusion with custom T2I weights.

**Image-to-Video Generation** The Image-to-Video (I2V) task can be viewed as a specialized variant of video prediction from individual frames (Babaeizadeh et al., 2017; Kim et al., 2019; Li et al., 2018). This task can be further classified into stochastic generation and conditional generation (Ni et al., 2023). Stochastic generation exclusively considers a given image, denoted as $x^0$ (Pan et al., 2019; Xiong et al., 2018; Li et al., 2018), while conditional generation takes into account additional conditions, such as text prompts, optical flow, or human pose (Kim et al., 2019).

In the realm of diffusion, LFDM (Ni et al., 2023) and I2VGen-XL (Wang et al., 2023; Luo et al., 2023) stand as some of the few existing works addressing this challenge. VideoComposer (Wang et al., 2023) and CoDi (Tang et al., 2023b), as multimodality approaches, also assert their capability in I2V generation.

## 5 LIMITATIONS AND FUTURE WORK

**Error Accumulation when Recurrently Processing:** Generating long videos, especially when the sequence exceeds the length of training video clips, can result in the loss of objects and a decline in visual quality. We hypothesize that this issue arises due to the accumulation of errors, particularly when employing a local sliding window approach. To mitigate this problem, increasing the look-back depth might be a potential solution. Additionally, incorporating skip connections or residual connections in the model architecture could aid in information propagation.

**Enhancing Look-back Modules:** Currently, our look-back modules and adapters primarily utilize simple 2D or 3D convolutions for modeling. Future research could explore the use of more complex look-back modules to enhance the model's expressive capacity. This could involve incorporating attention mechanisms, hidden states in look-back kernel, or other advanced techniques to improve the model's ability to capture long-range dependencies in video sequences.

## 6 CONCLUSION

Video serves as a digital record, capturing the dynamic visual intricacies of the real world. Video generation requires modeling the physical phenomena of the real world, which is more suitable to be viewed as a time series forecasting problem rather than a mere generative task, especially for Image-to-Video tasks.

In this work, we introduce a novel implicit semi-auto-regressive approach in diffusion U-Net to solve the I2V challenge of diffusion models. We propose look-back module, together with a hybrid strategy to initialize model input, with the purpose of enhancing the consistency with the given conditioned image. Experiments substantiate that our method excels in generating video sequences well-conditioned by a single image, producing more realistic and temporally coherent results than previous methods.

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

# A APPENDIX

## A.1 DIVERSE GENERATION RESULTS

We demonstrate the diverse generation results of our model here in Fig. 8.

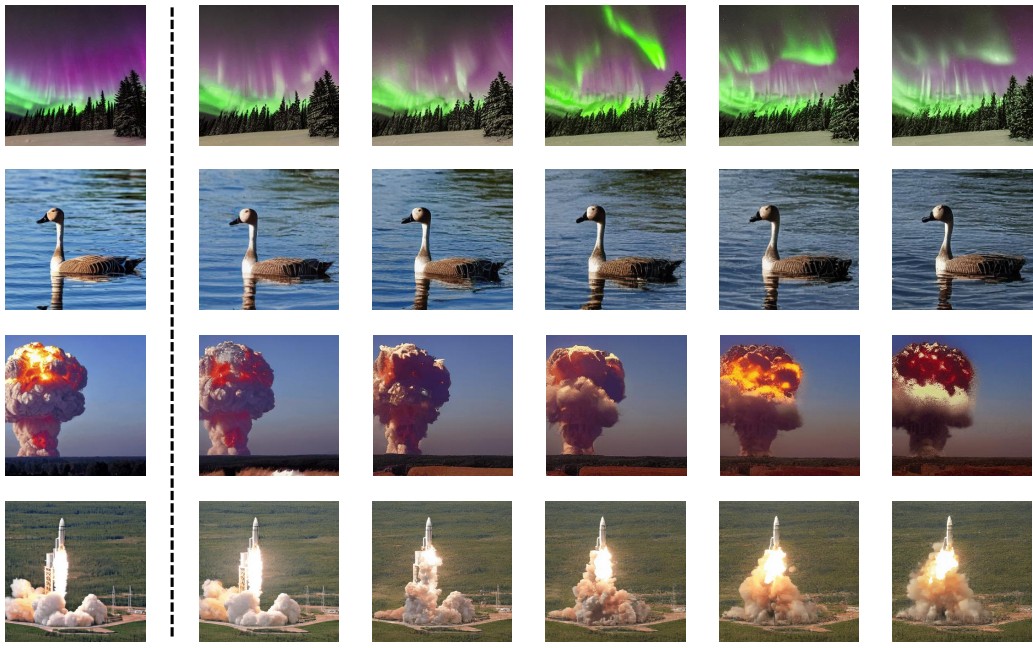

Figure 8: Diverse generation results.

## A.2 ABLATION OF DIFFERENT TEMPORAL MODELING METHODS

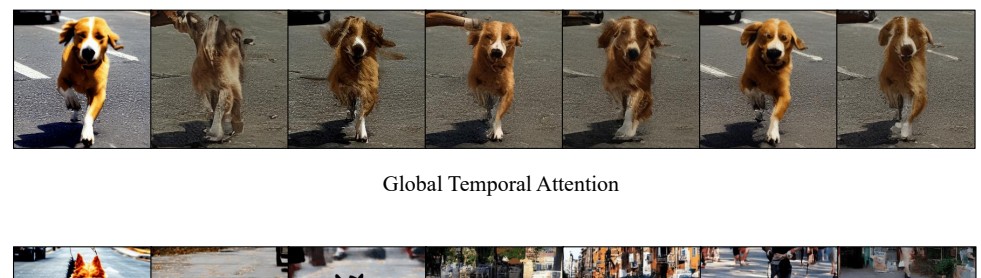

Global Temporal Attention

Masked Temporal Attention

Figure 9: More ablation results.

**Global Attention** We assess the widely employed global temporal attention in the I2V task. However, the outcomes exhibit inconsistency with the provided image. Fixing the first frame to the given image introduces a noticeable skip and flicker, detracting from the desired visual coherence.

**Masked Attention** An intuitive approach involves the use of masked temporal attention, employing an attention mask shaped as an upper-triangular matrix. This design ensures that each frame solely considers information from its preceding frames. Despite the simplicity of this method, its application in video, which inherently differs from language, often results in a lack of smooth transitions. Experimental findings also indicate its inadequacy in producing satisfactory results, leading to a compromise in temporal coherence across the entire video sequence with a even more information loss.

### A.3 GROUND TRUTH SIMILARITY CURVE BETWEEN EACH FRAME AND 1ST FRAME

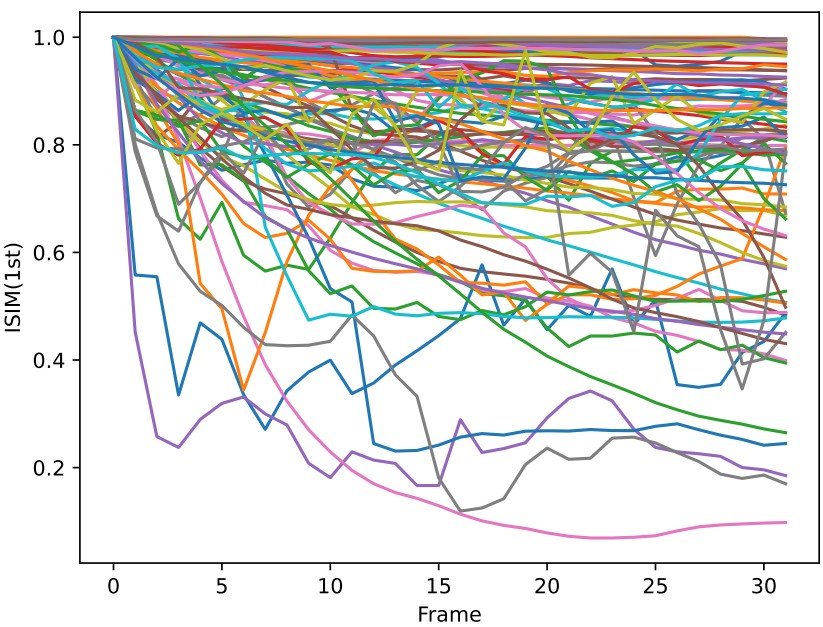

Figure 10: Results obtained from 100 videos sourced from WebVid-10M, each with a clip length of 32 frames. The majority of videos maintain a high similarity with the initial frame, although some outliers deviate from this pattern