# OpenReview forum: "Implicit Semi-auto-regressive Image-to-Video Diffusion"
_ICLR.cc/2024/Conference — Submitted to ICLR 2024_

### Official Review · Reviewer_hgs2 · 2023-10-31

**Soundness:** 2 fair
**Presentation:** 3 good
**Contribution:** 2 fair
**Rating:** 5
**Confidence:** 4

**Summary:**

This work proposes a new image-to-video framework with the main contribution coming from a new module called the temporal recurrent look-back approach. However, I believe this paper has significant issues in terms of completeness, including the choice of baselines and the unsatisfactory results in the experiments. Therefore, unless I am misunderstanding something, I am inclined to reject this paper.

**Strengths:**

+: The writing is quite clear.

**Weaknesses:**

- I find the setting of this paper somewhat strange. From my perspective, the results presented in the paper looks like video prediction, which should be compared with many video prediction baselines.

- The choice of baselines for comparison is confusing. Firstly, I don't understand why video prediction is not compared. Secondly, even if we exclude commercial software like Runway, there are other options for image-to-video, such as the results showcased on the official website of Make-a-Video. Additionally, tools like VideoCrafter and references [1] and [2] can also perform similar tasks.

- Does the proposed framework generate different results? For the same input image, the paper only shows one result.

- The algorithm's performance is not good. For example, in Figure 4, the third row shows a completely blurred face. Would this be considered a failure case? Furthermore, in Figure 5, it is evident that the predicted images have different brightness compared to the original images.

[1] MCVD: Masked Conditional Video Diffusion for Prediction, Generation, and Interpolation

[2] Diffusion Models for Video Prediction and Infilling

**Questions:**

-In fact, I don't understand why new metrics like ISIM were designed. Couldn't existing metrics like LPIPS be directly used for evaluation? What advantages does ISIM offer?

---

### Official Review · Reviewer_sTJV · 2023-10-31

**Soundness:** 1 poor
**Presentation:** 2 fair
**Contribution:** 2 fair
**Rating:** 3
**Confidence:** 5

**Summary:**

This paper presents a diffusion model for image-to-video generation. The method starts with a pre-trained text-to-image diffusion model. By inserting a look-back module and a 3D convolution, the inflated diffusion model can generate a video given the input condition. To generate a video given the first frame, the method concatenates the clean first frame with noisy subsequent frames as input to the diffusion model. As shown in the paper, the proposed model can generate subsequent frames that are relevant to the first frame.

**Strengths:**

- As shown in Figure 7, The look-back module is effective.
- By discarding the temporal attention, the computational cost is largely reduced when the number of frames is high.

**Weaknesses:**

- The idea of concatenating conditions with input noise volume is not novel in diffusion models. It has been used in many papers such as VideoComposer. The baselines (add noise or inversion) are weak, they are widely used in training-free diffusion models, but the proposed method is not a training-free model.
- In Figure 5, the results of I2VGen-XL and VideoComposer (especially VideoComposer) are much worse than the results shown in their paper. It will be good to explain the difference in the paper, i.e. what implementation of VideoComposer is being used in the comparison.
- In Figure-4, the temporal consistency / structure preservation is not very good.
- In terms of quantitative evaluation, only frame similarity and text-image alignment are measured. It will be helpful to provide other important metrics such as temporal consistency and image quality.

**Questions:**

How is the new look-back module initialized?

---

### Official Review · Reviewer_PWSR · 2023-11-01

**Soundness:** 3 good
**Presentation:** 2 fair
**Contribution:** 2 fair
**Rating:** 5
**Confidence:** 4

**Summary:**

This paper proposes a new approach to the Image-to-Video generation problem. The proposed method is based on a semi-autoregressive diffusion model. Specifically, the diffusion model’s UNet was extended with a novel temporal layer implemented with a 2D convolution-based windowed recurrent look-back mechanism. To enable the model to generate videos from a provided input image, the authors incorporate the image explicitly into the UNet input, along with the noisy version of the other frames to be generated during the denoising process of the diffusion model. Results from the experiments demonstrate that such model and input design enable the proposed system to generate plausible videos that are consistent with the input image.

**Strengths:**

This paper focuses on the Image-to-Video setting with diffusion models, which is an important research topic that is relatively under-explored compared to the Text-to-Video or Video-to-Video setting. Preserving the consistency with the input image is the unique challenge of this problem, for which this paper contributes interesting model and system design. In particular, the use of the temporal recurrent look-back mechanism for first-frame propagation is technically sound and interesting.

**Weaknesses:**

Lack of discussion and comparison with most relevant works on the I2V setting (e.g. [1], [2]). Although  I2V is relatively less explored than T2V, there have been works that were shown effective for the I2V tasks. While [1] and [2] focused mainly on T2V, their method can be naturally applied to perform I2V. They can generate a video that is highly consistent with the single input image. More discussions and comparisons with those existing works are needed to make it clearer for which aspects the proposed method offers more advantages compared to the current stage of knowledge on this problem.

It is not entirely clear from the current writing what the main advantages of the proposed temporal layers (based on 2DConv recurrent look-back) are compared to existing approaches to temporal modelling (such as 3DConv, ConvLSTM, or temporal attention). The authors had some limited discussion on this in the appendix, but no insight was provided, and very few visual examples. I believe this point needs more in-depth discussion and analysis because the proposed 2DConv look-back module is one of the key claimed contributions and novelties of this paper.
The paper in its current form left me with the impression that the experiments were not carried out to the extent that they should be, which makes the current results somewhat premature.
+ The authors mentioned that the model was trained with only 5000 clips and trained with only 72 hours on a single GPU. They mentioned that further training can lead to better performance but it is not clear how much and in which way it will improve.
 + I observed that the quality of the generated videos is generally worse than the existing general video generation works [2], [3]. Is this an indication of a limitation of using the 2DConv look-back module compared to more powerful temporal layers? Or is it the effect of the model being under-trained?
 + There are still noticeable jumps in appearance between the first and second frames, will training more make this better or worse?

Overall, while I appreciate the important problem that this paper proposes to solve, I found the current results unimpressive compared to existing methods that can solve the same problem. More importantly, such unsatisfactory performance could come from pre-mature experiments. I believe this paper can benefit from more thorough experiments and in-depth study.

[1] Make-A-Video: Text-to-Video Generation without Text-Video Data. Singer et al., 2022

[2] Align your Latents: High-Resolution Video Synthesis with Latent Diffusion Models. Blattmann et al., 2023

[3] Magic Video: Efficient Video Generation with Latent Diffusion Models. Zhou et al., 2022

**Questions:**

Please find my detailed comments in the Weaknesses section above.

---

### Official Review · Reviewer_7B9z · 2023-11-04

**Soundness:** 2 fair
**Presentation:** 3 good
**Contribution:** 2 fair
**Rating:** 5
**Confidence:** 3

**Summary:**

This paper proposes a temporal recurrent look-back approach to model video dynamics, which leverages prior information from the first frame (provided as a given image) as an implicit semi-auto-regressive process. A hybrid input initialization strategy is further introduced to enhance the propagation of information within the look-back module. Experiments on a subset of WebVid-2M demonstrate that the effectiveness of the proposed method on video detail consistency.

**Strengths:**

+ A look-back recurrent convolutional module inserted to diffusion U-Net is proposed to provide coherent generation results with high computation efficiency.

+ The generated results seem to be more consistent than other methods by leveraging prior information from the first frame as an implicit semi-auto-regressive process.

+ Some ablation studies are provided to facilitate the understanding of how the performance benefits from different components, including look-back module, different hybrid strategies and text involvement when inference.

**Weaknesses:**

- What does the “implicit” in the title mean? The paper does not clearly explain its meaning in the method section.

- The results of the proposed method are more consistent than those of other methods, however, some cases appear to lack dynamics, e.g., the second and third rows of Fig. 5.

- Some details of generated results are still inconsistent although the overall appearance looks similar, e.g., for the first row of Fig. 5, the colors of dog legs in the last two frames are different from the colors in the given frames.

- How do the number of previous layer hidden states L and the look-back depth d affect the performance? Some related parameters analysis should be conducted in the experimental section.

**Questions:**

1. What does the “implicit” in the title mean? How is the “implicit” reflected in the proposed framework?

2.  Some results appear to lack dynamics, e.g., the second and third rows of Fig. 5.

3. Some details of generated results are still inconsistent, e.g., for the first row of Fig. 5, the colors of dog legs in the last two frames are different from the colors in the given frames.

4. How do the number of previous layer hidden states L and the look-back depth d affect the performance?

---

### Meta-Review · Area_Chair_HwNN · 2023-12-01

**Metareview:**

The paper explores image-to-video generation, adding a temporal element to a diffusion UNet, a relatively under-explored problem area. However, it lacks rigor, particularly in methodology comparison and evaluation metrics. The quality of generations looks rather weak, and not convincingly demonstrated. The absence of a rebuttal to reviewers' comments further weakens the submission. Given these, I recommend rejection. I encourage the authors to thoroughly revise and address these issues for future submissions.

**Justification For Why Not Higher Score:**

The area addressed by the paper is not entirely under-explored; it's relatively less explored compared to text-to-video or video-to-video. Nonetheless, the execution of the paper lacks sufficient rigor. As raised by the reviewers:
- The results are weak.
- There are missing comparisons, ablations, and questions regarding evaluation metrics.
- The authors did not provide a rebuttal to address the reviewers' comments.

**Justification For Why Not Lower Score:**

N/A

---

### Decision · Program_Chairs · 2024-01-16

Reject